# No-Regret Learning in Bayesian Games

**Jason Hartline**
Northwestern University
Evanston, IL
hartline@northwestern.edu

**Vasilis Syrgkanis**
Microsoft Research
New York, NY
vasy@microsoft.com

**Éva Tardos**
Cornell University
Ithaca, NY
eva@cs.cornell.edu

## Abstract

Recent price-of-anarchy analyses of games of complete information suggest that coarse correlated equilibria, which characterize outcomes resulting from no-regret learning dynamics, have near-optimal welfare. This work provides two main technical results that lift this conclusion to games of incomplete information, a.k.a., Bayesian games. First, near-optimal welfare in Bayesian games follows directly from the smoothness-based proof of near-optimal welfare in the same game when the private information is public. Second, no-regret learning dynamics converge to Bayesian coarse correlated equilibrium in these incomplete information games. These results are enabled by interpretation of a Bayesian game as a stochastic game of complete information.

## 1 Introduction

A recent confluence of results from game theory and learning theory gives a simple explanation for why good outcomes in large families of strategically-complex games can be expected. The advance comes from (a) a relaxation the classical notion of equilibrium in games to one that corresponds to the outcome attained when players' behavior ensures asymptotic *no-regret*, e.g., via standard online learning algorithms such as *weighted majority*, and (b) an extension theorem that shows that the standard approach for bounding the quality of classical equilibria automatically implies the same bounds on the quality of no-regret equilibria. This paper generalizes these results from static games to Bayesian games, for example, auctions.

Our motivation for considering learning outcomes in Bayesian games is the following. Many important games model repeated interactions between an uncertain set of participants. Sponsored search, and more generally, online ad-auction market places, are important examples of such games. Platforms are running millions of auctions, with each individual auction slightly different and of only very small value, but such market places have high enough volume to be the financial basis of large industries. This online auction environment is best modeled by a repeated Bayesian game: the auction game is repeated over time, with the set of participants slightly different each time, depending on many factors from budgets of the players to subtle differences in the opportunities.

A canonical example to which our methods apply is a single-item first-price auction with players' values for the item drawn from a product distribution. In such an auction, players simultaneously submit sealed bids and the player with the highest bid wins and pays her bid. The utility of the winner is her value minus her bid; the utilities of the losers are zero. When the values are drawn from non-identical continuous distributions the Bayes-Nash equilibrium is given by a differential equation

that is not generally analytically tractable, cf. [8] (and generalizations of this model, computationally hard, see [3]). Again, though their Bayes-Nash equilibria are complex, we show that good outcomes can be expected in these kinds of auctions.

Our approach to proving that good equilibria can be expected in repeated Bayesian games is to extend an analogous result for static games,[1] i.e., the setting where the same game with the same payoffs and the same players is repeated. Nash equilibrium is the classical model of equilibrium for each stage of the static game. In such an equilibrium the strategies of players may be randomized; however, the randomizations of the players are independent. To measure the quality of outcomes in games Koutsoupias and Papadimitriou [9] introduced the *price of anarchy*, the ratio of the quality of the worst Nash equilibrium over a socially optimal solution. Price of anarchy results have been shown for large families of games, with a focus on those relevant for computer networks. Roughgarden [11] identified the canonical approach for bounding the price of anarchy of a game as showing that it satisfies a natural *smoothness* condition.

There are two fundamental flaws with Nash equilibrium as a description of strategic behavior. First, computing a Nash equilibrium can be PPAD hard and, thus, neither should efficient algorithms for computing a Nash equilibrium be expected nor should any dynamics (of players with bounded computational capabilities) converge to a Nash equilibrium. Second, natural behavior tends to introduce correlations in strategies and therefore does not converge to Nash equilibrium even in the limit. Both of these issues can be resolved for large families of games. First, there are relaxations of Nash equilibrium which allow for correlation in the players' strategies. Of these, this paper will focus on *coarse correlated equilibrium* which requires the expected payoff of a player for the correlated strategy be no worse than the expected payoff of any action at the player's disposal. Second, it was proven by Blum et al. [2] that the (asymptotic) no-regret property of many online learning algorithms implies convergence to the set of coarse correlated equilibria.[2]

Blum et al. [2] extended the definition of the price of anarchy to outcomes obtained when each player follows a no-regret learning algorithm.[3] As coarse correlated equilibrium generalize Nash equilibrium it could be that the worst case equilibrium under the former is worse than the latter. Roughgarden [11], however, observed that there is often no degradation; specifically, the very same smoothness property that he identified as implying good welfare in Nash equilibrium also proves good welfare of coarse correlated equilibrium (equivalently: for outcomes from no-regret learners). Thus, for a large family of static games, we can expect strategic behavior to lead to good outcomes.

This paper extends this theory to Bayesian games. Our contribution is two-fold: (i) We show an analog of the convergence of no-regret learning to coarse correlated equilibria in Bayesian games, which is of interest independently of our price of anarchy analysis; and (ii) we show that the coarse correlated equilibria of the Bayesian version of any smooth static game have good welfare. Combining these results, we conclude that no-regret learning in smooth Bayesian games achieves good welfare.

These results are obtained as follows. It is possible to view a Bayesian game as a stochastic game, i.e., where the payoff structure is fixed but there is a random action on the part of Nature. This viewpoint applied to the above auction example considers a population of bidders associated for each player and, in each stage, Nature uniformly at random selects one bidder from each population to participate in the auction. We re-interpret and strengthen a result of Syrgkanis and Tardos [12] by showing that the smoothness property of the static game (for any fixed profile of bidder values) implies smoothness of this stochastic game. From the perspective of coarse correlated equilibrium, there is no difference between a stochastic game and the non-stochastic game with each random variable replaced with its expected value. Thus, the smoothness framework of Roughgarden [11] extends this result to imply that the coarse correlated equilibria of the stochastic game are good. To show that we can expect good outcomes in Bayesian games, it suffices to show that no-regret learning converges to the coarse correlated equilibrium of this stochastic game. Importantly, when we consider learning algorithms there is a distinction between the stochastic game where players' payoffs are random variables and the non-stochastic game where players' payoffs are the expectation

of these variables. Our analysis addressed this distinction and, in particular, shows that, in the stochastic game on populations, no-regret learning converges almost surely to the set of coarse correlated equilibrium. This result implies that the average welfare of no-regret dynamics will be good, almost surely, and not only in expectation over the random draws of Nature.

## 2 Preliminaries

This section describes a general game theoretic environment which includes auctions and resource allocation mechanisms. For this general environment we review the results from the literature for analyzing the social welfare that arises from no-regret learning dynamics in repeated game play. The subsequent sections of the paper will generalize this model and these results to Bayesian games, a.k.a., games of incomplete information.

**General Game Form.** A general game $\mathcal{M}$ is specified by a mapping from a profile $a \in \mathcal{A} \equiv \mathcal{A}_1 \times \cdots \times \mathcal{A}_n$ of allowable actions of players to an outcome. Behavior in a game may result in (possibly correlated) randomized actions $\mathbf{a} \in \Delta(\mathcal{A})$.[4] Player $i$'s utility in this game is determined by a profile of individual values $v \in \mathcal{V} \equiv \mathcal{V}_1 \times \cdots \times \mathcal{V}_n$ and the (implicit) outcome of the game; it is denoted $U_i(\mathbf{a}; v_i) = \mathbb{E}_{a \sim \mathbf{a}} [U_i(a; v_i)]$. In games with a social planner or principal who does not take an action in the game, the utility of the principal is $R(\mathbf{a}) = \mathbb{E}_{a \sim \mathbf{a}} [R(a)]$. In many games of interest, such as auctions or allocation mechanisms, the utility of the principal is the revenue from payments from the players. We will use the term *mechanism* and *game* interchangeably.

In a *static game* the payoffs of the players (given by $v$) are fixed. Subsequent sections will consider *Bayesian games* in the independent private value model, i.e., where player $i$'s value $v_i$ is drawn independently from the other players' values and is known only privately to player $i$. Classical game theory assumes *complete information* for static games, i.e., that $v$ is known, and *incomplete information* in Bayesian games, i.e., that the distribution over $\mathcal{V}$ is known. For our study of learning in games no assumptions of knowledge are made; however, to connect to the classical literature we will use its terminology of complete and incomplete information to refer to static and Bayesian games, respectively.

**Social Welfare.** We will be interested in analyzing the quality of the outcome of the game as defined by the social welfare, which is the sum of the utilities of the players and the principal. We will denote by $SW(\mathbf{a}; v) = \sum_{i \in [n]} U_i(\mathbf{a}; v_i) + R(\mathbf{a})$ the expected social welfare of mechanism $\mathcal{M}$ under a randomized action profile $\mathbf{a}$. For any valuation profile $v \in \mathcal{V}$ we will denote the optimal social welfare, i.e, the maximum over outcomes of the game of the sum of utilities, by $\text{OPT}(v)$.

**No-regret Learning and Coarse Correlated Equilibria.** For complete information games, i.e., fixed valuation profile $v$, Blum et al. [2] analyzed repeated play of players using no-regret learning algorithms, and showed that this play converges to a relaxation of Nash equilibrium, namely, coarse correlated equilibrium.

**Definition 1** (no regret). *A player achieves* no regret *in a sequence of play* $a^1, \ldots, a^T$ *if his regret against any fixed strategy* $a'_i$ *vanishes to zero:*

$$\lim_{T \to \infty} \frac{1}{T} \sum_{t=1}^{T} (U_i(a'_i, a^t_{-i}; v_i) - U_i(a^t; v_i)) = 0. \tag{1}$$

**Definition 2** (coarse correlated equilibrium, CCE). *A randomized action profile* $\mathbf{a} \in \Delta(\mathcal{A})$ *is a* coarse correlated equilibrium *of a complete information game with valuation profile* $v$ *if for every player* $i$ *and* $a'_i \in \mathcal{A}_i$*:*

$$\mathbb{E}_{\mathbf{a}} [U_i(\mathbf{a}; v_i)] \geq \mathbb{E}_{\mathbf{a}} [U_i(a'_i, \mathbf{a}_{-i}; v_i)] \tag{2}$$

**Theorem 3** (Blum et al. [2]). *The empirical distribution of actions of any no-regret sequence in a repeated game converges to the set of* CCE *of the static game.*

**Price of Anarchy of CCE.** Roughgarden [11] gave a unifying framework for comparing the social welfare, under various equilibrium notions including coarse correlated equilibrium, to the optimal social welfare by defining the notion of a smooth game. This framework was extended to games like auctions and allocation mechanisms by Syrgkanis and Tardos [12].

| Game/Mechanism | $(\lambda, \mu)$ | POA | Reference |
|---|---|---|---|
| Simultaneous First Price Auction with Submodular Bidders | $(1 - 1/e, 1)$ | $\frac{e}{e-1}$ | [12] |
| First Price Multi-Unit Auction | $(1 - 1/e, 1)$ | $\frac{e}{e-1}$ | [5] |
| First Price Position Auction | $(1/2, 1)$ | $2$ | [12] |
| All-Pay Auction | $(1/2, 1)$ | $2$ | [12] |
| Greedy Combinatorial Auction with $d$-complements | $(1 - 1/e, d)$ | $\frac{de}{e-1}$ | [10] |
| Proportional Bandwitdth Allocation Mechanism | $(1/4, 1)$ | $4$ | [12] |
| Submodular Welfare Games | $(1, 1)$ | $2$ | [13, 11] |
| Congestion Games with Linear Delays | $(5/3, 1/3)$ | $5/2$ | [11] |

Figure 1: Examples of smooth games and mechanisms

**Definition 4** (smooth mechanism). *A mechanism $\mathcal{M}$ is $(\lambda, \mu)$-smooth for some $\lambda, \mu \geq 0$ there exists an independent randomized action profile $\mathbf{a}^*(v) \in \Delta(\mathcal{A}_1) \times \cdots \times \Delta(\mathcal{A}_n)$ for each valuation profile $v$, such that for any action profile $a \in \mathcal{A}$ and valuation profile $v \in \mathcal{V}$:*

$$\sum_{i \in [n]} U_i(\mathbf{a}_i^*(v), a_{-i}; v_i) \geq \lambda \cdot \text{OPT}(v) - \mu \cdot R(a). \tag{3}$$

Many important games and mechanisms satisfy this smoothness definition for various parameters of $\lambda$ and $\mu$ (see Figure 1); the following theorem shows that the welfare of any coarse correlated equilibrium in any of these games is nearly optimal.

**Theorem 5** (efficiency of CCE; [12]). *If a mechanism is $(\lambda, \mu)$-smooth then the social welfare of any course correlated equilibrium at least $\frac{\lambda}{\max\{1, \mu\}}$ of the optimal welfare, i.e., the price of anarchy satisfies* POA $\leq \frac{\max\{1, \mu\}}{\lambda}$.

**Price of Anarchy of No-regret Learning.** Following Blum et al. [2], Theorem 3 and Theorem 5 imply that no-regret learning dynamics have near-optimal social welfare.

**Corollary 6** (efficiency of no-regret dyhamics; [12]). *If a mechanism is $(\lambda, \mu)$-smooth then the average welfare of any no-regret dynamics of the repeated game with a fixed player set and valuation profile, achieves average social welfare at least $\frac{\lambda}{\max\{1, \mu\}}$ of the optimal welfare, i.e., the price of anarchy satisfies* POA $\leq \frac{\max\{1, \mu\}}{\lambda}$.

Importantly, Corollary 6 holds the valuation profile $v \in \mathcal{V}$ fixed throughout the repeated game play. The main contribution of this paper is in extending this theory to games of incomplete information, e.g., where the values of the players are drawn at random in each round of game play.

## 3 Population Interpretation of Bayesian Games

In the standard *independent private value model* of a *Bayesian game* there are $n$ players. Player $i$ has type $\mathbf{v}_i$ drawn uniformly from the set of type $\mathcal{V}_i$ (and this distribution is denoted $\mathcal{F}_i$).[5] We will restrict attention to the case when the type space $\mathcal{V}_i$ is finite. A player's strategy in this Bayesian game is a mapping $s_i : \mathcal{V}_i \to \mathcal{A}_i$ from a valuation $v_i \in \mathcal{V}_i$ to an action $a_i \in \mathcal{A}_i$. We will denote with $\Sigma_i = \mathcal{A}_i^{\mathcal{V}_i}$ the strategy space of each player and with $\Sigma = \Sigma_1 \times \ldots \times \Sigma_n$. In the game, each player $i$ realizes his type $v_i$ from the distribution and then makes action $s_i(v_i)$ in the game.

In the population interpretation of the Bayesian game, also called the *agent normal form representation* [6], there are $n$ finite populations of players. Each player in population $i$ has a type $v_i$ which we assume to be distinct for each player in each population and across populations.[6] The set of players in the population is denoted $\mathcal{V}_i$. and the player in population $i$ with type $v_i$ is called player $v_i$. In the population game, each player $v_i$ chooses an action $s_i(v_i)$. Nature uniformly draws one player from

each population, and the game is played with those players' actions. In other words, the utility of player $v_i$ from population $i$ is:

$$U_{i,v_i}^{\mathrm{AG}}(s) = \mathbb{E}_{\mathbf{v}}\left[U_i(s(\mathbf{v}); \mathbf{v}_i) \cdot 1\{\mathbf{v}_i = v_i\}\right] \tag{4}$$

Notice that the population interpretation of the Bayesian game is in fact a stochastic game of complete information.

There are multiple generalizations of coarse correlated equilibria from games of complete information to games of incomplete information (c.f. [6], [1], [4]). One of the canonical definitions is simply the coarse correlated equilibrium of the stochastic game of complete information that is defined by the population interpretation above.[7]

**Definition 7** (Bayesian coarse correlated equilibrium - BAYES-CCE). *A randomized strategy profile* $\mathbf{s} \in \Delta(\Sigma)$ *is a Bayesian coarse correlated equilibrium if for every* $a_i' \in A_i$ *and for every* $v_i \in \mathcal{V}_i$:

$$\mathbb{E}_{\mathbf{s}}\mathbb{E}_{\mathbf{v}}\left[U_i(\mathbf{s}(\mathbf{v}); \mathbf{v}_i) \mid \mathbf{v}_i = v_i\right] \geq \mathbb{E}_{\mathbf{s}}\mathbb{E}_{\mathbf{v}}\left[U_i(a_i', \mathbf{s}_{-i}(\mathbf{v}_{-i}); \mathbf{v}_i) \mid \mathbf{v}_i = v_i\right] \tag{5}$$

In a game of incomplete information the welfare in equilibrium will be compared to the expected ex-post optimal social welfare $\mathbb{E}_{\mathbf{v}}[\mathrm{OPT}(\mathbf{v})]$. We will refer to the worst-case ratio of the expected optimal social welfare over the expected social welfare of any BAYES-CCE as BAYES-CCE-POA.

## 4 Learning in Repeated Bayesian Game

Consider a repeated version of the population interpretation of a Bayesian game. At each iteration one player $v_i$ from each population is sampled uniformly and independently from other populations. The set of chosen players then participate in an instance of a mechanism $\mathcal{M}$. We assume that each player $v_i \in \mathcal{V}_i$, uses some no-regret learning rule to play in this repeated game.[8] In Definition 8, we describe the structure of the game and our notation more elaborately.

**Definition 8.** *The* repeated Bayesian game of $\mathcal{M}$ *proceeds as follows. In stage* $t$:

1. *Each player* $v_i \in \mathcal{V}_i$ *in each population* $i$ *picks an action* $s_i^t(v_i) \in A_i$. *We denote with* $s_i^t \in \mathcal{A}_i^{|\mathcal{V}_i|}$ *the function that maps a player* $v_i \in \mathcal{V}_i$ *to his action.*

2. *From each population* $i$ *one player* $v_i^t \in \mathcal{V}_i$ *is selected uniformly at random. Let* $v^t = (v_1^t, \ldots, v_n^t)$ *be the chosen profile of players and* $s^t(v^t) = (s_1^t(v_1^t), \ldots, s_n^t(v_n^t))$ *be the profile of chosen actions.*

3. *Each player* $v_i^t$ *participates in an instance of game* $\mathcal{M}$, *in the role of player* $i \in [n]$, *with action* $s_i^t(v_i^t)$ *and experiences a utility of* $U_i(s^t(v^t); v_i^t)$. *All players not selected in Step 2 experience zero utility.*

**Remark.** We point out that for each player in a population to achieve no-regret he does not need to know the distribution of values in other populations. There exist algorithms that can achieve the no-regret property and simply require an oracle that returns the utility of a player at each iteration. Thus all we need to assume is that each player receives as feedback his utility at each iteration. ∎

**Remark.** We also note that our results would extend to the case where at each period multiple matchings are sampled independently and players potentially participate in more than one instance of the mechanism $\mathcal{M}$ and potentially with different players from the remaining population. The only thing that the players need to observe in such a setting is their average utility that resulted from their action $s_i^t(v_i) \in \mathcal{A}_i$ from all the instances that they participated at the given period. Such a scenario seems an appealing model in online ad auction marketplaces where players receive only average utility feedback from their bids. ∎

**Bayesian Price of Anarchy for No-regret Learners.** In this repeated game setting we want to compare the average social welfare of any sequence of play where each player uses a vanishing regret algorithm versus the average optimal welfare. Moreover, we want to quantify the worst-case such average welfare over all possible valuation distributions within each population:

$$\sup_{\mathcal{F}_1,\ldots,\mathcal{F}_n} \limsup_{T\to\infty} \frac{\sum_{t=1}^{T}\text{OPT}(v^t)}{\sum_{t=1}^{T}SW^{\mathcal{M}}(s^t(v^t);v^t)} \tag{6}$$

We will refer to this quantity as the *Bayesian price of anarchy for no-regret learners*. The numerator of this term is simply the average optimal welfare when players from each population are drawn independently in each stage; it converges almost surely to the expected ex-post optimal welfare $\mathbb{E}_{\mathbf{v}}[\text{OPT}(\mathbf{v})]$ of the stage game. Our main theorem is that if the mechanism is smooth and players follow no-regret strategies then the expected welfare is guaranteed to be close to the optimal welfare.

**Theorem 9** (Main Theorem). *If a mechanism is $(\lambda, \mu)$-smooth then the average (over time) welfare of any no-regret dynamics of the repeated Bayesian game achieves average social welfare at least $\frac{\lambda}{\max\{1,\mu\}}$ of the average optimal welfare, i.e.* $\text{POA} \leq \frac{\max\{1,\mu\}}{\lambda}$, *almost surely.*

**Roadmap of the proof.** In Section 5, we show that any vanishing regret sequence of play of the repeated Bayesian game, will converge *almost surely* to the Bayesian version of a coarse correlated equilibrium of the incomplete information stage game. Therefore the Bayesian price of total anarchy will be upper bounded by the efficiency of guarantee of any Bayesian coarse correlated equilibrium. Finally, in Section 6 we show that the price of anarchy bound of smooth mechanisms directly extends to Bayesian coarse correlated equilibria, thereby providing an upper bound on the Bayesian price of total anarchy of the repeated game.

**Remark.** We point out that our definition of BAYES-CCE is inherently different and more restricted than the one defined in Caragiannis et al. [4]. There, a BAYES-CCE is defined as a joint distribution $D$ over $\mathcal{V} \times \mathcal{A}$, such that if $(\mathbf{v}, \mathbf{a}) \sim D$ then for any $v_i \in \mathcal{V}_i$ and $a_i'(v_i) \in \mathcal{A}_i$:

$$\mathbb{E}_{(\mathbf{v},\mathbf{a})}\left[U_i(\mathbf{a}; v_i)\right] \geq \mathbb{E}_{(\mathbf{v},\mathbf{a})}\left[U_i(a_i'(\mathbf{v}), \mathbf{a}_{-i}; v_i)\right] \tag{7}$$

The main difference is that the product distribution defined by a distribution in $\Delta(\Sigma)$ and the distribution of values, cannot produce any possible joint distribution over $(\mathcal{V}, \mathcal{A})$, but the type of joint distributions are restricted to satisfy a conditional independence property described by [6]. Namely that player $i$'s action is conditionally independent of some other player $j$'s value, given player $i$'s type. Such a conditional independence property is essential for the guarantees that we will present in this work to extend to a BAYES-CCE and hence do not seem to extend to the notion given in [4]. However, as we will show in Section 5, the no-regret dynamics that we analyze, which are mathematically equivalent to the dynamics in [4], do converge to this smaller set of BAYES-CCE that we define and for which our efficiency guarantees will extend. This extra convergence property is not needed when the mechanism satisfies the stronger *semi-smoothness* property defined in [4] and thereby was not needed to show efficiency bounds in their setting. ∎

## 5 Convergence of Bayesian No-Regret to BAYES-CCE

In this section we show that no-regret learning in the repeated Bayesian game converges almost surely to the set of Bayesian coarse correlated equilibria. Any given sequence of play of the repeated Bayesian game, which we defined in Definition 8, gives rise to a sequence of strategy-value pairs $(s^t, v^t)$ where $s^t = (s_1^t, \ldots, s_n^t)$ and $s_i^t \in \mathcal{A}_i^{\mathcal{V}_i}$, captures the actions that each player $v_i$ in population $i$ would have chosen, had they been picked. Then observe that all that matters to compute the average social welfare of the game for any given time step $T$, is the empirical distribution of pairs $(s, v)$, up till time step $T$, denoted as $D^T$, i.e. if $(\mathbf{s}^T, \mathbf{v}^T)$ is a random sample from $D^T$:

$$\tfrac{1}{T}\sum_{t=1}^{T} SW(s^t(v^t);v^t) = \mathbb{E}_{(\mathbf{s}^T,\mathbf{v}^T)}\left[SW(\mathbf{s}^T(\mathbf{v}^T);\mathbf{v}^T)\right] \tag{8}$$

**Lemma 10** (Almost sure convergence to BAYES-CCE). *Consider a sequence of play of the random matching game, where each player uses a vanishing regret algorithm and let $D^T$ be the empirical distribution of (strategy, valuation) profile pairs up till time step $T$. Consider any subsequence of $\{D^T\}_T$ that converges in distribution to some distribution $D$. Then, almost surely, $D$ is a product distribution, i.e. $D = D_s \times D_v$, with $D_s \in \Delta(\Sigma)$ and $D_v \times \Delta(\mathcal{V})$ such that $D_v = \mathcal{F}$ and $D_s \in$ BAYES-CCE of the static incomplete information game with distributional beliefs $\mathcal{F}$.*

*Proof.* We will denote with

$$r_i(a_i^*, a; v_i) = U_i(a_i^*, a_{-i}; v_i) - U_i(a; v_i),$$

the regret of player $v_i$ from population $i$, for action $a_i^*$ at action profile $a$. For a $v_i \in \mathcal{V}_i$ let $x_i^t(v_i) = \mathbf{1}\{v_i^t = v_i\}$. Since the sequence has vanishing regret for each player $v_i$ in population $P_i$, it must be that for any $s_i^* \in \Sigma_i$:

$$\sum_{t=1}^T x_i^t(v_i) \cdot r_i\left(s_i^*(v_i), s^t(v^t); v_i\right) \le o(T) \tag{9}$$

For any fixed $T$, let $D_s^T \in \Delta(\Sigma)$ denote the empirical distribution of $s^t$ and let $\mathbf{s}$ be a random sample from $D_s^T$. For each $s \in \Sigma$, let $\mathcal{T}_s \subset [T]$ denote the time steps such that $s^t = s$ for each $t \in \mathcal{T}_s$. Then we can re-write Equation (9) as:

$$\mathbb{E}_{\mathbf{s}}\left[\frac{1}{|\mathcal{T}_{\mathbf{s}}|} \sum_{t \in \mathcal{T}_{\mathbf{s}}} x_i^t(v_i) \cdot r_i\left(s_i^*(v_i), s^t(v^t); v_i\right)\right] \le \frac{o(T)}{T} \tag{10}$$

For any $s \in \Sigma$ and $w \in \mathcal{V}$, let $\mathcal{T}_{s,w} = \{t \in \mathcal{T}_s : v^t = w\}$. Then we can re-write Equation (10) as:

$$\mathbb{E}_{\mathbf{s}}\left[\sum_{w \in \mathcal{V}} \frac{|\mathcal{T}_{s,w}|}{|\mathcal{T}_{\mathbf{s}}|} \mathbf{1}\{w_i = v_i\} \cdot r_i\left(s_i^*(v_i), \mathbf{s}(w); v_i\right)\right] \le \frac{o(T)}{T} \tag{11}$$

Now we observe that $\frac{|\mathcal{T}_{s,w}|}{|\mathcal{T}_s|}$ is the empirical frequency of the valuation vector $w \in \mathcal{V}$, when filtered at time steps where the strategy vector was $s$. Since at each time step $t$ the valuation vector $v^t$ is picked independently from the distribution of valuation profiles $\mathcal{F}$, this is the empirical frequency of $\mathcal{T}_s$ independent samples from $\mathcal{F}$.

By standard arguments from empirical processes theory, if $\mathcal{T}_s \to \infty$ then this empirical distribution converges almost surely to the distribution $\mathcal{F}$. On the other hand if $\mathcal{T}_s$ doesn't go to $\infty$, then the empirical frequency of strategy $s$ vanishes to $0$ as $T \to \infty$ and therefore has measure zero in the above expectation as $T \to \infty$. Thus for any convergent subsequence of $\{D^T\}$, if $D$ is the limit distribution, then if $s$ is in the support of $D$, then almost surely the distribution of $w$ conditional on strategy $s$ is $\mathcal{F}$. Thus we can write $D$ as a product distribution $D_s \times \mathcal{F}$.

Moreover, if we denote with $\mathbf{w}$ the random variable that follows distribution $\mathcal{F}$, then the limit of Equation (11) for any convergent sub-sequence, will give that:

$$\text{a.s.: } \mathbb{E}_{\mathbf{s} \sim D_s} \mathbb{E}_{\mathbf{w} \sim \mathcal{F}}\left[\mathbf{1}\{\mathbf{w}_i = v_i\} \cdot r_i\left(s_i^*(v_i), \mathbf{s}(\mathbf{w}); v_i\right)\right] \le 0$$

Equivalently, we get that $D_s$ will satisfy that for all $v_i \in \mathcal{V}_i$ and for all $s_i^*$:

$$\text{a.s.: } \mathbb{E}_{\mathbf{s} \sim D_s} \mathbb{E}_{\mathbf{w} \sim \mathcal{F}}\left[r_i\left(s_i^*(\mathbf{w}_i), \mathbf{s}(\mathbf{w}); \mathbf{w}_i\right) \mid \mathbf{w}_i = v_i\right] \le 0$$

The latter is exactly the BAYES-CCE condition from Definition 7. Thus $D_s$ is in the set of BAYES-CCE of the static incomplete incomplete information game among $n$ players, where the type profile is drawn from $\mathcal{F}$. ∎

Given the latter convergence theorem we can easily conclude the following the following theorem, whose proof is given in the supplementary material.

**Theorem 11.** *The price of anarchy for Bayesian no-regret dynamics is upper bounded by the price of anarchy of Bayesian coarse correlated equilibria, almost surely.*

## 6   Efficiency of Smooth Mechanisms at Bayes Coarse Correlated Equilibria

In this section we show that smoothness of a mechanism $\mathcal{M}$ implies that any BAYES-CCE of the incomplete information setting achieves at least $\frac{\lambda}{\max\{1,\mu\}}$ of the expected optimal welfare. To show this we will adopt the interpretation of BAYES-CCE that we used in the previous section, as coarse correlated equilibria of a more complex normal form game; the stochastic agent normal form representation of the Bayesian game. We can interpret this complex normal form game as the game that arises from a complete information mechanism $\mathcal{M}^{\text{AG}}$ among $\sum_i |\mathcal{V}_i|$ players, which randomly samples one player from each of the $n$ population and where the utility of a player in the complete information mechanism $\mathcal{M}^{\text{AG}}$ is given by Equation (4). The set of possible outcomes in this agent

game corresponds to the set of mappings from a profile of chosen players to an outcome in the underlying mechanism $\mathcal{M}$. The optimal welfare of this game, is then the expected ex-post optimal welfare $\mathrm{OPT}^{\mathrm{AG}} = \mathbb{E}_{\mathbf{v}}\left[\mathrm{OPT}(\mathbf{v})\right]$.

The main theorem that we will show is that whenever mechanism $\mathcal{M}$ is $(\lambda, \mu)$-smooth, then also mechanism $\mathcal{M}^{\mathrm{AG}}$ is $(\lambda, \mu)$-smooth. Then we will invoke a theorem of [12, 11], which shows that any coarse correlated equilibrium of a complete information mechanism achieves at least $\frac{\lambda}{\max\{1,\mu\}}$ of the optimal welfare. By the equivalence between BAYES-CCE and CCE of this complete information game, we get that every BAYES-CCE of the Bayesian game achieves at least $\frac{\lambda}{\max\{1,\mu\}}$ of the expected optimal welfare.

**Theorem 12** (From complete information to Bayesian smoothness)**.** *If a mechanism $\mathcal{M}$ is $(\lambda, \mu)$-smooth, then for any vector of independent valuation distributions $\mathcal{F} = (\mathcal{F}_1, \ldots, \mathcal{F}_n)$, the complete information mechanism $\mathcal{M}^{\mathrm{AG}}$ is also $(\lambda, \mu)$-smooth.*

*Proof.* Consider the following randomized deviation for each player $v_i \in \mathcal{V}_i$ in population $i$: He random samples a valuation profile $\mathbf{w} \sim \mathcal{F}$. Then he plays according to the randomized action $\mathbf{s}_i^*(v_i, \mathbf{w}_{-i})$, i.e., the player deviates using the randomized action guaranteed by the smoothness property of mechanism $\mathcal{M}$ for his type $v_i$ and the random sample of the types of the others $\mathbf{w}_{-i}$.

Consider an arbitrary action profile $s = (s_1, \ldots, s_n)$ for all players in all populations. In this context it is better to think of each $s_i$ as a $|\mathcal{V}_i|$ dimensional vector in $\mathcal{A}_i^{|\mathcal{V}_i|}$ and to view $s$ as a $\sum_i |\mathcal{V}_i|$ dimensional vector. Then with $s_{-v_i}$ we will denote all the components of this large vector except the ones corresponding to player $v_i \in \mathcal{V}_i$. Moreover, we will be denoting with $\mathbf{v}$ a sample from $\mathcal{F}$ drawn by mechanism $\mathcal{M}^{\mathrm{AG}}$. We now argue about the expected utility of player $v_i$ from this deviation, which is:

$$\mathbb{E}_{\mathbf{w}}\left[U_{i,v_i}^{\mathrm{AG}}(s_i^*(v_i, \mathbf{w}_{-i}), s_{-v_i})\right] = \mathbb{E}_{\mathbf{w}}\mathbb{E}_{\mathbf{v}}\left[U_i(s_i^*(v_i, \mathbf{w}_{-i}), s_{-i}(\mathbf{v}_{-i}); v_i) \cdot \mathbb{1}\{\mathbf{v}_i = v_i\}\right]$$

Summing the latter over all players $v_i \in \mathcal{V}_i$ in population $i$:

$$\sum_{v_i \in \mathcal{V}_i} \mathbb{E}_{\mathbf{w}}\left[U_{i,v_i}^{\mathrm{AG}}(s_i^*(v_i, \mathbf{w}_{-i}), s_{-v_i})\right] = \mathbb{E}_{\mathbf{w},\mathbf{v}}\left[\sum_{v_i \in \mathcal{V}_i} U_i(s_i^*(v_i, \mathbf{w}_{-i}), s_{-i}(\mathbf{v}_{-i}); v_i) \cdot \mathbb{1}\{\mathbf{v}_i = v_i\}\right]$$

$$= \mathbb{E}_{\mathbf{v},\mathbf{w}}\left[U_i(\mathbf{s}_i^*(\mathbf{v}_i, \mathbf{w}_{-i}), s_{-i}(\mathbf{v}_{-i}); \mathbf{v}_i)\right]$$
$$= \mathbb{E}_{\mathbf{v},\mathbf{w}}\left[U_i(\mathbf{s}_i^*(\mathbf{w}_i, \mathbf{w}_{-i}), s_{-i}(\mathbf{v}_{-i}); \mathbf{w}_i)\right]$$
$$= \mathbb{E}_{\mathbf{v},\mathbf{w}}\left[U_i(\mathbf{s}_i^*(\mathbf{w}), s_{-i}(\mathbf{v}_{-i}); \mathbf{w}_i)\right],$$

where the second to last equation is an exchange of variable names and regrouping using independence. Summing over populations and using smoothness of $\mathcal{M}$, we get smoothness of $\mathcal{M}^{\mathrm{AG}}$:

$$\sum_{i \in [n]} \sum_{v_i \in \mathcal{V}_i} \mathbb{E}_{\mathbf{w}}\left[U_{i,v_i}^{\mathrm{AG}}(s_i^*(v_i, \mathbf{w}_{-i}), s_{-v_i})\right] = \mathbb{E}_{\mathbf{v},\mathbf{w}}\left[\sum_{i \in [n]} U_i(\mathbf{s}_i^*(\mathbf{w}), s_{-i}(\mathbf{v}_{-i}); \mathbf{w}_i)\right]$$
$$\geq \mathbb{E}_{\mathbf{v},\mathbf{w}}\left[\lambda \mathrm{OPT}(\mathbf{w}) - \mu R(s(\mathbf{v}))\right] = \lambda \mathbb{E}_{\mathbf{w}}\left[\mathrm{OPT}(\mathbf{w})\right] - \mu R^{AG}(s)$$

■

**Corollary 13.** *Every* BAYES-CCE *of the incomplete information setting of a smooth mechanism $\mathcal{M}$, achieves expected welfare at least $\frac{\lambda}{\max\{1,\mu\}}$ of the expected optimal welfare.*

## 7 Finite Time Analysis and Convergence Rates

In the previous section we argued about the limit average efficiency of the game as time goes to infinity. In this section we analyze the convergence rate to BAYES-CCE and we show approximate efficiency results even for finite time, when players are allowed to have some $\epsilon$-regret.

**Theorem 14.** *Consider the repeated matching game with a $(\lambda, \mu)$-smooth mechanism. Suppose that for any $T \geq T^0$, each player in each of the $n$ populations has regret at most $\frac{\epsilon}{n}$. Then for every $\delta$ and $\rho$, there exists a $T^*(\delta, \rho)$, such that for any $T \geq \min\{T^0, T^*\}$, with probability $1 - \rho$:*

$$\frac{1}{T}\sum_{t=1}^{T} SW(s^t(v^t); v^t) \geq \frac{\lambda}{\max\{1,\mu\}}\mathbb{E}_{\mathbf{v}}\left[\mathrm{OPT}(\mathbf{v})\right] - \delta - \mu \cdot \epsilon \qquad (12)$$

*Moreover, $T^*(\delta, \rho) \leq \frac{54 \cdot n^3 \cdot |\Sigma| \cdot |\mathcal{V}|^2 \cdot H^3}{\delta^3} \log\left(\frac{2}{\rho}\right)$.*

## Footnotes

[1]In the standard terms of the game theory literature, we extend results for learning in games of complete information to games of incomplete information.

[2]This result is a generalization of one of Foster and Vohra [7].

[3]They referred to this price of anarchy for no-regret learners as the *price of total anarchy*.

[4]Bold-face symbols denote random variables.

[5]The restriction to the uniform distribution is without loss of generality for any finite type space and for any distribution over the type space that involves only rational probabilities.

[6]The restriction to distinct types is without of loss of generality as we can always augment a type space with an index that does not affect player utilities.

[7]This notion is the coarse analog of the *agent normal form Bayes correlated equilibrium* defined in Section 4.2 of Forges [6].

[8]An equivalent and standard way to view a Bayesian game is that each player draws his value independently from his distribution each time the game is played. In this interpretation the player plays by choosing a strategy that maps his value to an action (or distribution over actions). In this interpretation our no-regret condition requires that the player not regret his actions for each possible value.

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
