[Supplementary Material · bayes-no-regret-supp.pdf]

# Supplementary material for "No-Regret Learning in Bayesian Games"

## A  Proof of Theorem 11

For readability we repeat the definitions of Lemma 10 and Theorem 11 from the main text.

**Lemma 10**  *Let $D \in \Delta(\Sigma \times \mathcal{V})$ be a joint distribution of (strategy, valuation) profile pairs. Consider a sequence of play of the random matching game, where each player uses a vanishing regret algorithm and let $D^T$ be the empirical distribution of strategy, valuation profile pairs up till time step $T$. Suppose that there exists a subsequence of $\{D^T\}_T$ that converges in distribution to $D$. Then, almost surely, $D$ is a product distribution, i.e. $D = D_s \times D_v$, with $D_s \in \Delta(\Sigma)$ and $D_v \times \Delta(\mathcal{V})$ such that $D_v = \mathcal{F}$ and $D_s \in$ BAYES-CCE of the static incomplete information game with distributional beliefs $\mathcal{F}$.*

**Theorem 11**  *The price of anarchy for Bayesian no-regret dynamics is upper bounded by the price of anarchy of Bayesian coarse correlated equilibria.*

*Proof.* Let $D \in \Delta(\Sigma \times \mathcal{V})$ be a joint distribution, such that there is a subsequence of $\{D^T\}_T$, converging in distribution $D$. Then by Lemma 10, almost surely, $D$ is a product distribution, i.e. $D \in \Delta(\Sigma) \times \Delta(\mathcal{V})$ and that the marginal on $\mathcal{V}$ is equal to $\mathcal{F}$ and the marginal on $\Sigma$ is a BAYES-CCE of the static incomplete information game with distributional beliefs $\mathcal{F}$.

Therefore, if $\rho$ is the BAYES-CCE $-$ POA of the mechanism, and if $(\mathbf{s}, \mathbf{v})$ is a random sample from $D$, then almost surely:

$$\mathbb{E}_{\mathbf{s},\mathbf{v}}\left[SW(\mathbf{s}(\mathbf{v}); \mathbf{v})\right] \geq \frac{1}{\rho}\mathbb{E}_{\mathbf{v}}\left[\text{OPT}(\mathbf{v})\right] \tag{13}$$

Thus the limit average social welfare of any convergent subsequence will be at least $\frac{1}{\rho}\mathbb{E}_{\mathbf{v}}\left[\text{OPT}(\mathbf{v})\right]$, which then implies that almost surely:

$$\lim_{T\to\infty} \inf \frac{1}{T}\sum_{t=1}^{T} SW(s^t(v^t); v^t) \geq \frac{1}{\rho}\mathbb{E}_{\mathbf{v}}\left[\text{OPT}(\mathbf{v})\right] = \frac{1}{\rho}\lim_{T\to\infty}\frac{1}{T}\sum_{t=1}^{T}\text{OPT}(v^t)$$

Thus for any non-measure zero event, for any $\epsilon$, there exists a $f(\epsilon)$ such that for any $T \geq f(\epsilon)$:

$$\frac{1}{T}\sum_{t=1}^{T} SW(s^t(v^t); v^t) \geq \frac{1}{\rho}\frac{1}{T}\sum_{t=1}^{T}\text{OPT}(v^t) - \epsilon$$

With no loss of generality we can assume that $\mathbb{E}_{\mathbf{v}}\left[\text{OPT}(\mathbf{v})\right] > 0$ (o.w. valuations are all zero and theorem holds trivially). Since, the average optimal welfare converges almost surely to $\mathbb{E}_{\mathbf{v}}\left[\text{OPT}(\mathbf{v})\right]$, we get that for any non-measure zero event, there exists a $g(\delta)$ such that for $T \geq g(\delta)$, $\frac{1}{T}\sum_{t=1}^{T}\text{OPT}(v^t)$ is bounded away from zero. Thereby, we can turn the additive error into a multiplicative one, i.e. for any non-measure zero event and for any $\epsilon'$ there exists $w(\epsilon')$ such that for any $T \geq w(\epsilon')$:

$$\frac{1}{T}\sum_{t=1}^{T} SW(s^t(v^t); v^t) \geq \frac{1}{\rho}(1+\epsilon')\frac{1}{T}\sum_{t=1}^{T}\text{OPT}(v^t)$$

This implies that almost surely:

$$\lim_{T\to\infty} \sup \frac{\frac{1}{T}\sum_{t=1}^{T}\text{OPT}(v^t)}{\frac{1}{T}\sum_{t=1}^{T} SW(s^t(v^t); v^t)} \leq \rho = \text{BAYES-CCE-POA}$$

■

# B  Proof of Theorem 14

**Theorem 14** *Consider the repeated matching game with a $(\lambda, \mu)$-smooth mechanism. Suppose that for any $T \geq T^0$, each player in each of the $n$ populations has regret at most $\frac{\epsilon}{n}$. Then for every $\delta$ and $\rho$, there exists a $T^*(\delta, \rho)$, such that for any $T \geq \min\{T^0, T^*\}$, with probability $1 - \rho$:*

$$\frac{1}{T} \sum_{t=1}^{T} SW(s^t(v^t); v^t) \geq \frac{\lambda}{\max\{1, \mu\}} \mathbb{E}_{\mathbf{v}}\left[\text{OPT}(\mathbf{v})\right] - \delta - \mu \cdot \epsilon \tag{14}$$

*Moreover, $T^*(\delta, \rho) \leq \frac{54 \cdot n^3 \cdot |\Sigma| \cdot |\mathcal{V}|^2 \cdot H^3}{\delta^3} \log\left(\frac{2}{\rho}\right)$.*

*Proof.* Fix a population $i$ and a Bayesian strategy $s_i^* \in \Sigma_i$, as well as a Bayesian strategy profile $s \in \Sigma$. For shorter notation we will denote:

$$\pi_i(s_i^*, s, v) = U_i(s_i^*(v_i), s_{-i}(v_{-i}); v_i).$$

For a time step $T$, let $p^T(s) = \frac{|\mathcal{T}_s|}{T}$ be the empirical distribution of a Bayesian strategy $s$ and with $p^T(v|s) = \frac{|\mathcal{T}_{s,v}|}{|\mathcal{T}_s|}$ be the empirical distribution of values conditional on a Bayesian strategy $s$. The average utility of a population $i$ up till time step $T$, when switching to a fixed Bayesian strategy $s_i^*$, can be written as:

$$\frac{1}{T} \sum_{t=1}^{T} \pi(s_i^*, s^t, v^t) = \sum_{s \in \Sigma} p^T(s) \sum_{v \in \mathcal{V}} p^T(v|s) \cdot \pi_i(s_i^*, s, v) \tag{15}$$

We will show that for any $s_i^*$, there exists a $T^*(\delta, \rho)$ such that for any $T \geq T^*(\delta, \rho)$, with probability $1 - \rho$:

$$\sum_{s \in \Sigma} p^T(s) \sum_{v \in \mathcal{V}} p^T(v|s) \cdot \pi_i(s_i^*, s, v) \geq \sum_{s \in \Sigma} p^T(s) \mathbb{E}_{\mathbf{v}}\left[\pi_i(s_i^*, s, \mathbf{v})\right] - \delta \tag{16}$$

where $\mathbf{v}$ is a random variable drawn from the distribution of valuation profiles $\mathcal{F}$. We will denote with $p(v)$ the density function implied by distribution $\mathcal{F}$.

In what follows we will denote with $H = \max_{i \in [n], v_i \in \mathcal{V}_i, x_i \in \mathcal{X}_i} v_i(x_i)$ the maximum possible value of any player. Thus observe that the utility of any player is upper bounded by $H$ and that the revenue collected by any player at equilibrium is upper bounded by $H$.

For a time period $T$, let $G = \{s \in \Sigma : p^T(s) \geq \zeta\}$. Then observe that:

$$\sum_{s \in \Sigma} p^T(s) \sum_{v \in \mathcal{V}} \left(p^T(v|s) - p(v)\right) \cdot \pi_i(s_i^*, s, v) \geq$$

$$\sum_{s \in G} p^T(s) \sum_{v \in V} \left(p^T(v|s) - p(v)\right) \cdot \pi_i(s_i^*, s, v) - \zeta \cdot |\Sigma| \cdot H$$

Observe that for any $s \in G$, $|\mathcal{T}_s| \geq \zeta \cdot T$. Thus $p^T(v|s)$ is the empirical mean of at least $\zeta \cdot T$ independent random samples of a Bernoulli trial with success probability $p(v)$. Hence, by Hoeffding bounds, we have that $|p^T(v|s) - p(v)| \leq t$ with probability at least $1 - 2\exp\left(-2 \cdot \zeta \cdot T \cdot t^2\right)$. Thus with that much probability we get:

$$\sum_{s \in \Sigma} p^T(s) \sum_{v \in \mathcal{V}} \left(p^T(v|s) - p(v)\right) \cdot \pi_i(s_i^*, s, v) \geq -t \cdot |\mathcal{V}| \cdot H - \zeta \cdot |\Sigma| \cdot H$$

By setting $t = \frac{\delta}{2 \cdot |\mathcal{V}| \cdot H}$, $\zeta = \frac{\delta}{2 \cdot |\Sigma| \cdot H}$ and $T^*(\delta, \rho) = \frac{16 \cdot |\Sigma| \cdot |\mathcal{V}|^2 \cdot H^3}{\delta^3} \log\left(\frac{2}{\rho}\right)$, we get the claimed property in Equation (16).

Now suppose that after time step $T^0$ each player in a population has regret $\epsilon/n$. Thus the average utility of the population is at least the utility from switching to any fixed Bayesian strategy $s_i^*$, minus an error term of $\epsilon/n$:

$$\sum_{s \in \Sigma} p^T(s) \sum_{v \in \mathcal{V}} p^T(v|s) \pi_i(s_i, s, v) \geq \sum_{s \in \Sigma} p^T(s) \sum_{v \in \mathcal{V}} p^T(v|s) \pi_i(s_i^*, s, v) - \frac{\epsilon}{n} \tag{17}$$

From the previous analysis, for any $T \geq \min\{T^0, T^*(\frac{2\delta}{3 \cdot n}, \rho)\}$, we get that with probability $1 - \rho$:

$$\sum_{s \in \Sigma} p^T(s) \sum_{v \in \mathcal{V}} p^T(v|s) \pi_i(s_i, s, v) \geq \sum_{s \in \Sigma} p^T(s) \mathbb{E}_{\mathbf{v}} \left[ \pi_i(s_i^*, s, \mathbf{v}) \right] - \frac{2\delta}{3n} - \frac{\epsilon}{n} \tag{18}$$

Summing over all populations and using the Bayesian smoothness property of the mechanism from Theorem 12, we have that with probability $1 - \rho$:

$$\sum_{s \in \Sigma} p^T(s) \sum_{v \in \mathcal{V}} p^T(v|s) \sum_i \pi_i(s_i, s, v) \geq \sum_{s \in \Sigma} p^T(s) \left( \lambda \mathbb{E}_{\mathbf{v}} \left[ \text{OPT}(\mathbf{v}) \right] - \mu R^{\text{AG}}(s) \right) - \frac{2\delta}{3} - \epsilon$$

$$\geq \lambda \mathbb{E}_{\mathbf{v}} \left[ \text{OPT}(\mathbf{v}) \right] - \mu \sum_{s \in \Sigma} p^T(s) R^{\text{AG}}(s) - \frac{2\delta}{3} - \epsilon$$

To conclude the theorem we observe that since for any $s \in \Sigma$, $|p^T(v|s) - p(v)| \leq \frac{\delta}{3 \cdot n \cdot |\mathcal{V}| \cdot H}$, we get that:

$$R^{\text{AG}}(s) = \sum_{v \in \mathcal{V}} p(v) R(s(v)) \leq \sum_{v \in \mathcal{V}} p^T(v|s) R(s(v)) + \frac{\delta}{3} \tag{19}$$

Since, the revenue collected by a player at any action in the support of an equilibrium is at most $H$. By the latter we can combine the revenue on the right hand side with the utility on the left hand side. We can also bound the remaining $(\mu - 1)$ of the revenue, by $(\mu - 1)$ of the average welfare minus $\epsilon$, since each player in each population can always drop out of the auction and therefore his average utility at an $\frac{\epsilon}{n}$-regret sequence must be at least $-\frac{\epsilon}{n}$.

Hence, we get that:

$$\sum_{s \in \Sigma} p^T(s) \sum_{v \in \mathcal{V}} p^T(v|s) SW(s(v); v) \geq \frac{\lambda}{\max\{1, \mu\}} \mathbb{E}_{\mathbf{v}} \left[ \text{OPT}(\mathbf{v}) \right] - \delta - \mu \cdot \epsilon \tag{20}$$

Thus choosing $T^*(\rho, \frac{2\delta}{3 \cdot n}) = \frac{54 \cdot n^3 \cdot |\Sigma| \cdot |\mathcal{V}|^2 \cdot H^3}{\delta^3} \log\left(\frac{2}{\rho}\right)$, we get the conditions of the theorem. ∎