[Reviews · NeurIPS 2015]

Submitted by Assigned_Reviewer_1

This paper attempts to show that the price of anarchy bounds already achieved via the smoothness property of games extends to results on the quality of learning outcomes in Bayesian games. To achieve that, the authors interpreted a Bayesian game as a stochastic game of complete information where there are a large population of possible participants. Once this assumption done, two important results are shown: First the outcome converges to a form of Bayesian corase correlated equilibrium when all players use a form of no-regret learning in the considered stochastic game. Second, the considedred stochatic game is also smooth if the game or the mechanism is itself smooth. These two reslts show that the price of anarchy results obtained via the smoothness technique extend to outcomes of no-regret learning in Bayesian games. This paper is very hard to read and to evaluate; it is more convenient for the "Game Theory" community than for the "NIPS" one (as its references reflect). Futhermore, it seems not be written specifically for the NIPS conference, it is just a "cut and past" from a extended version. This is reflected by the last section where the authors are saying: "In this section, we analyze the convergence..." and nothing is specified about this convergence. Some questions, remarks and suggestions: (a) why the extra material just gives proofs of Th 10 and 13? How about other Ths/Lemmas? (b) why the correlated equilibrium is this one of Forges? (c) the Th 11 is not clear, do you mean it converges to BCCE? please elaborate more on that. (d) Selfish players are hard to manage, except when we add a central entity as is the case via the correlated EQ. With this sort of equilibrium, the problem is more easy to solve (in general) but does this reflect a realistic approach?

Summary: The main result of this paper is to show that the price of anarchy bounds achieved via the smoothness property of games or mechanisms extends to results on the quality of learning outcomes in Bayesian games. The paper is more convenient for the "Game Theory" community than for the "NIPS" one.

Submitted by Assigned_Reviewer_2

In game theory, one can show that under certain smoothness assumptions, the so called price of anarchy of a full information game is bounded. The price of anarchy is the ratio of the optimal social welfare over the worst social welfare achievable under certain type of equilibrium. The literature on this subject is quite extensive and bounds on the price of anarchy have been shown for the large class of coarse correlated equilibria. In the world of games of incomplete information, however, these results do not exist and only bounds on the price of anarchy of Bayes-Nash equilibria exist.

The main contribution of this paper is to provide a bound on the price of anarchy for coarse correlated equilibria. The way this is done is by reducing these game to several full information games.

The paper is really hard to read and seems to lack generality. Even the setup was not completely clear to me. What do the authors mean by

"F_i be the empirical distributions of values in population P_i" which samples are being used to talk about the empirical distribution?

I think that the main issue of the paper comes from Lemma 10. The hypothesis of the lemma are not completely clear to me. First of all, they say that D^T is an empirical distribution of plays and they say if a subsequence of D^T converges in distribution to D then ... However, D^T is a random measure so is the convergence in distribution a.s? Must the subsequence be the same for all values of D^T? This same type of questions arise on equation (10) after taking the limit. In a way they are taking the limit of the form

E_{s \sim D^T}[f_T(s)]

and they claim that because f_T \right arrow f a.s. and D^T \rightarrow D in distribution (which again does not make much sense), then the above term will converge to E_{s \sim D} [f(s)]. It is not entirely clear to me that this will hold in general and if it does then the authors should provide a better explanation.

The rest of the paper seems to be correct so if the authors can address my comments on Lemma 10 I would consider accepting the paper. Nevertheless, I don't believe that the subject of the paper would be of much interest to the NIPS community.

Summary: OK paper that extends the price of anarchy results of full information games to Bayesian games.

The paper is really hard to read and I suspect that one of the proofs is wrong.

Author Feedback
Author rebuttal: We thank very much the reviewers for the comments, suggestions and references and address some of their questions below.

Reviewer 1:
(a) All the proofs of theorems in the paper are either in the supplementary or in the main body of the paper (except theorems in preliminaries which appear in previous work). We acknowledge that there is a small discrepancy in the appendix, and we apologize for any confusion: Due to separate compiling of the appendix and main body the numbering in the appendix is incorrect. Specifically, Theorem 10 in the appendix refers to what is Theorem 11 in the main text and Theorem 13 refers to what is theorem 14 in the main text. These are the only two Theorems that we don't have the proof in the main body of the paper. The proofs of the rest of the lemmas appear in the main body.
(c) In this theorem we argue that the welfare of the empirical distribution of no-regret dynamics converges to a welfare that is at least as large as the welfare of the worst Bayes correlated equilibrium. We are not arguing about convergence of the empirical distributions. We do invoke though a type of distributional convergence result that we proved in Lemma 10 to show this welfare statement. Specifically, by Lemma 10 we get that the empirical distribution of any convergent subsequence, converges almost surely to the set of BCCE (from Theorem 10), not necessarily to a single point in the set of BCCE. Even though the latter is only for any convergent subsequence, and only towards the set of BCCE, this is sufficient to show that the welfare of the empirical distribution must be at least the worst welfare of any BCCE. This is why we chose to phrase it as a price of anarchy statement, since welfare results on no-regret dynamics is our ultimate goal in the paper. The proof can also be found in the appendix, which might clear some of the confusion.
(d) Even though selfish players are hard to manage, our results in the paper and in the prior literature on efficiency of no-regret dynamics shows that if selfish players invoke natural learning algorithms with minimal no-regret guarantees, then the efficiency of the system is close to optimal. And this arises without some central entity enforcing the player actions or enforcing the exact learning algorithm that the players need to invoke.

Reviewer 2:
1) When we say that F_i is the empirical distribution of values in population P_i, we mean that if we select one player at random from population P_i and look at the value of this player, then the number that comes up from this random process if viewed as a random variable, has distribution F_i.
2) With respect to the technical comment about Lemma 10: in this lemma we fix the realization of the sequence of random measures D^1,..., D^T,... and hence we argue about convergence pointwise for each realization of the random measures D^T. Subsequently the conclusion holds of the lemma holds almost surely over the randomness of the realized distributions D^T. So in that respect the convergent subsequence that we refer to in the lemma doesn't need to be the same for different realized measures.
3) For the second objection with respect to convergence of the expectation of the summation in equation (10), we provide a formal argument below the equation. Potentially a missing point is that D^T is over a finite probability space. Then (borrowing the reviewers notation) we have D^T -> D and f_T->f a.s.. Then observe that because D^T is over a finite probability space, we can write the expectation as \sum_{s} D^T(s)*f_T(s). Now both D^T(s) -> D(s) and f_T(s) -> f(s), hence the product should also be converging to D(s)*f(s). Hence, also the finite sum of the products should be converging.

Review 5:
We omitted a final conclusion section due to space constraints, instead summarized the main conclusion of the paper as the closing of the introductions stating that " our main contribution is to show that the price of anarchy results obtained via the smoothness technique extend to outcomes of no-regret learning in Bayesian games."

Once again we thank all the reviewers for their feedback and will address all the comments in the final version of the paper.